# Development and Assessment of SNP Genotyping Arrays for Citrus and Its Close Relatives

**DOI:** 10.3390/plants13050691

**Published:** 2024-02-29

**Authors:** Yoko Hiraoka, Sergio Pietro Ferrante, Guohong Albert Wu, Claire T. Federici, Mikeal L. Roose

**Affiliations:** 1Department of Botany and Plant Sciences, University of California, Riverside, CA 92521, USA; yoko.eck@ucr.edu (Y.H.);; 2US Department of Energy Joint Genome Institute, Walnut Creek, CA 94598, USA

**Keywords:** *Citrus*, genotyping array, plant breeding, germplasm, single nucleotide polymorphism

## Abstract

Rapid advancements in technologies provide various tools to analyze fruit crop genomes to better understand genetic diversity and relationships and aid in breeding. Genome-wide single nucleotide polymorphism (SNP) genotyping arrays offer highly multiplexed assays at a relatively low cost per data point. We report the development and validation of 1.4M SNP Axiom^®^ Citrus HD Genotyping Array (Citrus 15AX 1 and Citrus 15AX 2) and 58K SNP Axiom^®^ Citrus Genotyping Arrays for *Citrus* and close relatives. SNPs represented were chosen from a citrus variant discovery panel consisting of 41 diverse whole-genome re-sequenced accessions of *Citrus* and close relatives, including eight progenitor citrus species. SNPs chosen mainly target putative genic regions of the genome and are accurately called in both *Citrus* and its closely related genera while providing good coverage of the nuclear and chloroplast genomes. Reproducibility of the arrays was nearly 100%, with a large majority of the SNPs classified as the most stringent class of markers, “*PolyHighResolution*” (PHR) polymorphisms. Concordance between SNP calls in sequence data and array data average 98%. Phylogenies generated with array data were similar to those with comparable sequence data and little affected by 3 to 5% genotyping error. Both arrays are publicly available.

## 1. Introduction

Citrus, a member of the family Rosaceae, is one of the most widely cultivated and economically valued fruit tree crops in the world. It is cultivated in more than 100 countries mostly in a belt that spans Mediterranean and subtropical climatic regions of the world [1], and its production is estimated at 143 million tons [2]. Citrus fruits are rich in various functional compounds such as vitamins, limonoids, and antioxidants and are nutritionally beneficial for human health [3,4]. *Citrus* is generally diploid and highly heterozygous, with a chromosome number of 2n = 18 and an estimated genome size of 360 Mb [5]. There are approximately 29,000 coding genes with an average sequence length of 1.3 kb and 5.8 exons [5]. Citrus has been domesticated and cultivated for several thousand years [6], and domestication-related selection by humans has tremendously shaped its genome. Modern *Citrus* varieties are thought to result from modifications of the genome guided by human selection. Domestication is thought to have started in Southeast Asia and dispersed to the rest of the world [6], leading to interspecific hybridization and introgression of ancestral progenitors into cultivated Citrus varieties. The exact lineages from ancestral *Citrus* to many of the modern cultivated Citrus types have not been resolved. Several features of Citrus biology and cultivation make deciphering these origins difficult [7]. Many species of *Citrus* hybridize sexually interspecifically and, occasionally, even intergenerically [8], resulting in highly heterozygous genomes such as in sweet orange [5,9]. Most commercially important citrus genotypes are either natural or artificial interspecific hybrids derived by hybridization among four major ancestral Citrus taxa: citron (*C. medica*), pummelo (*C. maxima*), mandarin (*C. reticulata*), and papeda (*C. micrantha*), which then further diversify by mutation and, in some cases, hybridization [9,10]. Citrus phylogeny studies are further complicated by the clonal propagation methods (grafting and asexual seed production) typically used to maintain desirable combinations of traits, narrowing the genetic diversity of many important cultivated groups. Diversity within these groups develops through the accumulation of somatic mutations, either as sport mutations or nucellar variants [7,11,12]. Many commercially important citrus, including navel oranges, Valencia oranges, Eureka lemons, Lisbon lemons, and clementines, are clonal groups.

Understanding the relationships between genomic variations and phenotypic variation associated with various horticultural traits is the key to the breeding of novel and improved citrus varieties. Molecular markers such as Simple Sequence Repeats (SSRs) and Amplified Fragment Length Polymorphism (AFLP) have previously been popular marker systems often utilized in citrus breeding and diversity studies for their large information content [13,14,15,16]. However, some previous marker systems were sequence anonymous, had low throughput, or did not provide enough density across the genome for high-resolution genomic studies in multiple populations. Recent advances in sequencing and genomic technologies enabled the construction of reference genomes of *C. sinensis* [5,17,18], *C. clementina* (or *C. reticulata*) [7], and *C. maxima* [19], facilitating high-resolution genomic studies and characterization of germplasm. However, the availability and utilization of genomic resources for citrus have lagged behind most of the major crops due to a lack of high-throughput genomic tools available and appropriate to analyze complex citrus populations.

In recent years, comprehensive genome-wide SNP polymorphisms are easily mined from sequence data and serve as genetic markers for GWAS [10], population structure analysis [20], and admixture analysis [21] in various populations, linkage disequilibrium (LD) analysis [10] and haplotype construction [7], as well as structural variant (SV)/copy number variation (CNV) and loss of heterozygosity (LOH) detection [22] in mutationally derived populations. High-throughput genotyping technologies, such as SNP genotyping, have revolutionized the way we can analyze the origin and genetic control of traits in crops. These tools are now available to crop breeders and produce high-fidelity data to perform high-resolution genomic studies, as demonstrated in apple, maize, rice, peach, and strawberry [23,24,25,26,27]. The development of SNP genotyping arrays requires a tremendous amount of knowledge about the organisms under investigation [28] to maximize the opportunities for SNP detection and to ensure that selected SNPs will be suitable for the final application of the arrays. Despite these prerequisites and the large initial investment, for some purposes, an array-based genotyping approach is a more efficient alternative to the whole genome sequencing approach [29,30] and the genotyping by sequencing (GBS) approach in terms of computational requirements necessary for downstream analysis for non-model organisms such as citrus. GBS utilizes restriction enzyme technologies to generate a representation (subset) of the genome that is then sequenced [31]. However, imputing missing calls can be computationally challenging and demands well-established complex pipelines to analyze a highly heterozygous citrus genome [32]. Fixed genotyping platforms such as arrays also facilitate the comparison of results among laboratories or germplasm collections compared to GBS methods. Therefore, array-based genotyping can provide a high-throughput and cost-effective technology. To provide citrus breeders with proper tools to rapidly characterize citrus germplasm, locate genes responsible for traits, and facilitate the rapid release of novel citrus varieties, we report here the development of two genome-wide high-density SNP genotyping arrays for citrus: Axiom^®^ Citrus HD Genotyping Array and Axiom^®^ Citrus Genotyping Array (1.4 million SNPs and 58,000 SNPs, respectively). The Citrus HD array was used to select a smaller subset of SNPs for the more economical Genotyping array but can also be used when increased SNP density is needed. In the present article, we also demonstrate the potential applications and utility of Axiom^®^ Citrus HD Genotyping Array and Axiom^®^ Citrus Genotyping Array by performing population structure analysis and phylogenetic analysis in diverse citrus germplasm and anticipate additional downstream analyses such as linkage mapping, GWAS, diversity analysis, CNV, and LOH analysis.

## 2. Results

### 2.1. Variant Selection According to In Silico Analysis of Re-Sequencing Data

This analysis identified 19.9M (19,907,622) autosomal variants and 2216 chloroplast variants from the whole genome sequence data of 41 diverse accessions of citrus species (33 accessions, Groups 1 and 2) and close relatives (8 accessions, Group 3) in the citrus variant discovery panel (Table 1 and Table 2). The 19.9 M sequence variants resulted from filtering according to the sequence quality and their support, as described in 4.2. Variant Selection According to In Silico Analysis of Re-Sequencing Data. In the 19.9M autosomal variants mined in the citrus variant discovery panel, the observed heterozygous calls ranged from 1.84% in Ethrog citron to 19.37% in Schaub rough lemon (Appendix A).

### 2.2. Axiom^®^ Citrus HD Genotyping Array

The Axiom^®^ Citrus HD Genotyping Array is comprised of Axiom^®^ Citrus HD Genotyping Array AX1 and Axiom^®^ Citrus HD Genotyping Array AX2 due to the number of SNPs and the Axiom^®^ design configuration. A total of 1,404,442 probes targeting 1,296,733 autosomal SNPs and 1002 chloroplast SNPs were selected to be represented after variant filtering part 1, followed by in silico analysis of candidate variants by the Affymetrix™ bioinformatics team (Santa Clara, CA, USA). SNP loci represented on the Axiom^®^ Citrus HD Genotyping Array included 854,128 SNPs from accession Group 1, 36,070 SNPs from Group 2, 285,521 SNPs from Group 3, 35,218 SNPs shared by Groups 1 and 2, 22,402 SNPs shared by Groups 1 and 3, 3517 SNPs shared by Groups 2 and 3 and finally 59,877 SNPs shared by all three groups (Appendix A). The chromosomal distribution of the SNPs represented on the Axiom^®^ Citrus HD Genotyping Array (Table 3a, Appendix A) and heterozygous SNPs in selected genotypes were assessed to ensure adequate genome coverage. Axiom^®^ Citrus HD Genotyping Array AX1 contains a total of 702,244 probes targeting 648,969 SNPs, of which 7882 are duplicate probes tiled on both strands targeting 3941 SNPs. A total of 254 genomic DNA samples, of which 232 were unique, were analyzed on the Axiom^®^ Citrus HD Genotyping Array AX1, of which 220 samples (86.61%) passed and 34 samples failed. A total of 362,281 SNPs were categorized as PHR (51.59%) and passed Affymetrix™ SNP filtering metrics. For these SNPs, the average cluster call rate (CR), average technical reproducibility (defined as an analysis of the same DNA sample), and average clonal reproducibility (defined as an analysis of DNA samples from separate trees of the same accession) were 99.52%, 99.86%, and 99.80%, respectively. The Axiom^®^ Citrus HD Genotyping Array AX2 contains a total of 702,198 probes targeting 648,766 SNPs, of which 8090 are duplicate probes tiled on both strands targeting 4045 SNPs. Of 254 genomic DNA samples that were analyzed on the Axiom^®^ Citrus HD Genotyping Array AX2, 224 samples (88.19%) passed, and 30 samples failed. A total of 366,550 SNPs were categorized as PHR (52.2%) and passed Affymetrix™ SNP filtering metrics. Average cluster CR, average technical reproducibility, and average clonal reproducibility were 99.56%, 99.80%, and 99.97%, respectively. A total of 240 samples without accessions belonging to Group 3 were analyzed; 353,278 SNPs were categorized as PHR (50.31%) on the Axiom^®^ Citrus HD Genotyping Array AX1, and 336,851 SNPs were categorized as PHR (47.97%) on the Axiom^®^ Citrus HD Genotyping Array AX2.

### 2.3. PHR Probes on the Axiom^®^ Citrus HD Genotyping Array AX1 and the Axiom^®^ Citrus HD Genotyping Array AX2

A total of 728,831 probes (50.55%) targeting 609,890 SNPs were categorized as PHR and passed Affymetrix™ SNP filtering metrics (Table 4, Figure 1a). The QC CR of the 34 samples that failed on the Axiom^®^ Citrus HD Genotyping Array AX1 and/or Array AX2 with Axiom^®^ Analysis Suite Default settings ranged from 67.46% to 96.99%. Out of 34 samples that failed, 20 samples were distant relatives, and 10 samples were *Poncirus* with an average QC CR of 94.18% and 95.46%, respectively. Three out of thirty-four failed samples had less than 80% QC CR and were distant *Citrus* relatives, *Murraya paniculata*, *Aegle marmelos*, and *Balsamocitrus daweii*. Five failed samples were synthetic polyploid hybrids, which were constructed by mixing DNA samples of two or more genotypes, which included some more distant relatives.

### 2.4. Axiom^®^ Citrus Genotyping Array

A total of 58,433 probes targeting 57,933 autosomal SNPs and 500 chloroplast SNPs were selected to be represented on the smaller Axiom^®^ Citrus Genotyping Array after variant filtering part 2. SNP loci represented on the Axiom^®^ Citrus Genotyping Array included 37,377 SNPs from (polymorphic in) Group 1, 2157 SNPs from Group 2, 8788 SNPs from Group 3, 2961 SNPs shared by Groups 1 and 2, 1454 SNPs shared by Groups 1 and 3, 198 SNPs shared by Groups 2 and 3, and finally 4998 SNPs shared by all three groups (Appendix A).

### 2.5. PHR Probes on the Axiom^®^ Citrus Genotyping Array

A total of 1507 genomic DNA samples were analyzed on the Axiom^®^ Citrus Genotyping Array; 1494 samples (99.14%) passed, and 101 samples failed (mostly polyploids, synthetic hybrids, or more distant relatives). A total of 51,296 probes (87.79%) targeting 51,296 SNPs were categorized as PHR and passed Affymetrix™ SNP filtering metrics (Table 4, Figure 1b). For 46 out of 50 CVC samples that failed on the Axiom^®^ Citrus Genotyping Array with Axiom^®^ Suite Default settings, the cluster CR ranged from 58.91% to 96.9%. Samples of four more distant citrus relatives, *Murraya paniculata*, *Clausena anisata*, *Clausena harmandiana*, and *Glycosmis pentaphylla*, had dQC values ranging from 0.657 to 0.787, and no Cluster CR was calculated. The 19 failed samples with less than 80% Cluster CR were all distant *Citrus* relatives. Average cluster CR, average technical reproducibility, and average clonal reproducibility were 99.60%, 99.75%, and 99.79%, respectively.

### 2.6. Chromosome Distribution and Functional Annotation of the Axiom^®^ Citrus HD Genotyping Array and the Axiom^®^ Citrus Genotyping Array Markers

Distributions of SNP markers along the Clementine 1.0 reference genome were assessed to determine whether there was adequate genome coverage for downstream analyses. SNP densities on the Axiom^®^ Citrus HD Genotyping Array and Axiom^®^ Citrus Genotyping Array were 3.9 SNPs/kb and 0.16 SNPs/kb, respectively (Table 4). The SNP distribution among chromosomes was roughly proportional to the genome length of each chromosome (Table 3a). The putative targets of probes represented on the Axiom^®^ Citrus HD Genotyping Array were assessed to show that 630,821 (49%) probes target 23,434 genes out of 24,533 Phytozome annotated genes, and on the Axiom^®^ Citrus Genotyping Array, 38,289 (66%) probes target 14,355 genes, as they were designed (Table 3b). Distribution of the SNPs on the Axiom^®^ Citrus HD Genotyping Array and the Axiom^®^ Citrus Genotyping Array along the genome was assessed by plotting the SNPs against Phytozome *C. clementina* v1.0 reference genome (Figure 2 and Appendix A). The SNPs are well distributed throughout the genome, with the largest marker gaps at 454,807 bases (chromosome 5) for the Axiom^®^ Citrus HD Genotyping Array and 953,089 bases (chromosome 5) for the Axiom^®^ Citrus Genotyping Array. In addition, SNPs on the Axiom^®^ Citrus HD Genotyping Array and the Axiom^®^ Citrus Genotyping Array were plotted against the gene density of Phytozome *C. clementina* v1.0 reference genome to show that the regions of high gene density correspond well to the regions of high SNP density (Figure 2).

### 2.7. Heterozygosity and Reproducibility of Genotyping Calls between Sequencing Data and the Axiom^®^ Citrus HD Genotyping Array

Percent average heterozygosity in the sequence data was estimated among the variants retained in the 19M VCF file generated from sequence data and separately for the array data (Figure 3). The highest estimated percent heterozygosity with both sequence data and array data was observed with Schaub rough lemon (JAM), and the lowest estimated percent heterozygosity was observed with Ethrog citron (ETH). Lower estimated percent heterozygosity was observed with array data in comparison to sequence data in *C. hystrix* (HYS), Flying Dragon trifoliate (FDT), Pomeroy trifoliate (PON), Yunnanese citron (YUN), *M. australasica* (AFL), *E. glauca* (ADL), *M. australis* (ARL), *C. micrantha* var. *microcarpa* (MIC), Buddha’s Hand citron (BUD) and Ethrog citron (ETH) due to SNP selection processes for the array design. Hybrid groups such as lemons (*C. limon*), limes (*C. aurantifolia*), and sweet oranges (*C. sinensis*) had higher heterozygosity with array data than the sequence data, and more distant groups such as papeda, trifoliate orange, *Microcitrus*, and *Eremocitrus* had lower heterozygosity with array data. The r² value between the estimated percent heterozygosity with sequence data and array data was 0.75. Heterozygosity in Mexican lime (MXL) was underestimated with the array data compared with sequence data and other accessions (Figure 3), probably because it is a papeda hybrid [33] and papeda-specific alleles are somewhat under-represented on the array. Percent average heterozygosity in selected taxa was estimated with SNP genotyping data of the Axiom^®^ Citrus HD Genotyping Array (Appendix A) and the Axiom^®^ Citrus Genotyping Array (Appendix A). The highest estimated percent average heterozygosity was observed in hybrid groups, such as sweet oranges (*C. sinensis*), lemons (*C. limon*), and limes (*C. aurantifolia*), and lowest in citron (*C. medica*) and the *Citrus* relative trifoliate orange (*P. trifoliata)* with data from both arrays. Estimated percent average heterozygosity of 1.4% and 2.6% in citron and trifoliate orange, respectively, with the Axiom^®^ Citrus Genotyping Array data are relatively low; however, in the case of trifoliate orange, there are still more than 4500 (Axiom^®^ Citrus HD Genotyping Array) and 1000 (Axiom^®^ Citrus Genotyping Array) heterozygous SNPs that can be expected to be useful in mapping traits in these taxa. In addition, the average percent heterozygosity values, estimated using genotyping data from the Axiom^®^ Citrus HD Genotyping Array and the Axiom^®^ Citrus Genotyping Array, were proportionally comparable to those measured from the sequencing data (Figure 3). Application of the Axiom^®^ Citrus Genotyping Array to citrus mapping can be estimated from the chromosomal distribution of heterozygous SNPs in various accessions. The distributions of heterozygous markers in accessions with high (sweet orange; 18,159 heterozygous markers), medium (Chandler pummelo; 4554 heterozygous markers), and low (Pomeroy trifoliate; 1293 heterozygous markers) heterozygosity (Figure 2) show that, even in *Poncirus*, maps with excellent coverage are predicted. Concordance between the Axiom^®^ Citrus Genotyping Array SNP data and sequence data of Flying Dragon trifoliate 3330A and Pomeroy trifoliate 1717 are 98.66% and 98.63%, respectively. Among the 632 loci at which calls did not match, 82.1% to 79.4% of calls were homozygous sites with sequence data, wrongly called as heterozygous with the Axiom^®^ Citrus Genotyping Array. Further investigation of the cluster plots with Axiom^®^ Analysis Suite showed that hybridization signals were significantly lower at these loci.

### 2.8. Genotyping Accuracy and Reproducibility of Genotyping Calls of Axiom^®^ Citrus HD Genotyping Array and Axiom^®^ Citrus Genotyping Array

The genotyping accuracy of the Axiom^®^ Citrus HD Genotyping Array and the Axiom^®^ Citrus Genotyping Array was assessed by comparing genotype calls for 635,809 PHR loci in 17 accessions and 51,296 PHR loci in 28 accessions from the citrus variant discovery panel that were also genotyped with these arrays. The pairwise concordance of a genotype between the two data sets was defined as the fraction of the SNP genotype calls for a given SNP that are in agreement with each other. “NoCall” was excluded from the calculation. The concordance between sequence data and Axiom^®^ Citrus HD Genotyping Array data ranged from 94.88% in Mexican lime to 99.57% in Sun Chu Sha mandarin, and the average concordance was 98.09% with a median of 98.93% (Figure 4). The reproducibility of genotyping data between the Axiom^®^ Citrus HD Genotyping Array and the Axiom^®^ Citrus Genotyping Array was assessed using 36 accessions that were analyzed with both arrays. Concordance of the genotyping data on the same accession analyzed by both arrays ranged from 99.14% for *C. amblycarpa* CRC 2485 to 99.84% for Royal grapefruit CRC 248, and the average concordance was 99.79% with a median of 99.87% (Figure 5).

### 2.9. Percent Identity of Accessions in Groups Composed of Clonal Accessions

The percent identity of accessions in groups composed of clonal accessions was assessed by comparison of genotype calls with the Genotyping array at 51,296 PHR loci within each group. Groups assessed included Navel orange (*C. sinensis*, 67 accessions), Valencia orange (*C. sinensis*, 25 accessions), Satsuma mandarin (*C. unshiu*, 45 accessions), Eureka and Lisbon lemon (24 accessions), and Clementine (*C. clementina*, 14 accessions), and percent identity in each group ranged from 97.94 to 100%, 99.60 to 100%, 99.08 to 100%, 99.08 to 99.98%, and 99.77 to 100%, respectively. The percent identity of accessions between Navel oranges and Valencia oranges ranged from 98.12 to 99.95%.

### 2.10. PCA

PCA was carried out with both the Axiom^®^ Citrus HD Genotyping Array and the Axiom^®^ Citrus Genotyping Array genotyping data. PCA with the Citrus HD Genotyping Array analyzed genotyping calls at 728,831 PHR loci of 196 diverse genomic DNA samples from the CVC (Appendix A). The first three principal components (PCs), PC1, PC2, and PC3, explained 62.2% (35.6%, 22.5%, and 4.1%, respectively) of the total variation in the dataset. The representatives in the citrus variant discovery panel appear at three of the corners of the 196 unique accessions plotted (Figure 6). PCA was also carried out with genotyping calls at 51,296 PHR loci for combined data of both arrays from 871 diploid CVC accessions (Appendix A, Figure 7). The first three PCs explained 59.7% (30.1%, 18.5%, and 11.1%, respectively) of the total variation in the dataset. PC4 explained 6.11% of the variation in the dataset. PCA carried out with these 871 accessions show that accessions in groups constituted of clonal accessions, *C. sinensis* (171 accessions), *C. unshiu* (45 accessions), *C. clementina* (14 accessions), and *Citrus limon* (L.) Burm. f. (Eureka and Lisbon lemons: 24 accessions) clustered tightly together, showing almost no variation within each group (Figure 7d). All three PCA analyses showed that three classical *Citrus* ancestral taxa [34,35], mandarins (*C. reticulata*), pummelos (*C. maxima*), and citrons (*C. medica*) form discrete, well-separated clusters. In addition, PC3 in all three PCA analyses is mostly explained by the inclusion of accessions of *Poncirus* (red) and its hybrids with *Citrus* (gray), kumquat (*Fortunella*) (pink), and papeda (yellow) in the data analyzed.

### 2.11. Phylogenetic Inference

Phylogeny was inferred for the variant discovery panel and selected CVC accessions. Analysis of the citrus variant discovery panel was performed separately for the Axiom^®^ Citrus Genotyping Array data and corresponding sequence data (Figure 8). A total of 42,987 and 32,754 LD-filtered SNPs were included for phylogenetic tree construction with Axiom^®^ Citrus Genotyping Array and sequence data (Figure 8), respectively. Sequence data included fewer SNPs than array data for this analysis due to more missing calls. The bootstrap values of above 90% are shown as bold lines. Both phylogenetic trees are supported with high bootstrap values, with 89% and 83% of the above 90% bootstrap values being 100% in the trees generated with array data and sequence data, respectively. A topological comparison of the trees performed with Phylo.io [36] showed that the trees displayed mostly similar relationships (dark blue) except for some branching patterns (light blue) (Appendix A). Analysis of 87 CVC accessions was performed using 41,626 LD-filtered SNPs from the Axiom^®^ Citrus Genotyping Array data. The bootstrap values of above 90% are shown in bold, and the phylogenetic tree is supported with high bootstrap values, with 89% of the above 90% bootstrap values corresponding to 100% bootstrap support (Figure 9). These 87 accessions were considered non-hybrid and non-admixed accessions and belonged to major *Citrus* taxa, *C. medica*, *C. maxima*, *C. reticulata*, and *Citrus* relatives, *Fortunella*, papeda, and outgroup genus *P. trifoliata.* The citrus cluster is well separated from outgroup *Poncirus* with 100 percent bootstrap support, and the five *Citrus* and *Fortunella* taxa form discrete clusters with complete monophyly and 100% bootstrap support. The accessions belonging to mandarin, pummelo, and citron are confined to three of the major clusters, and those belonging to *C. reticulata* appear to be more divergent within the cluster based on their branch length in comparison to other clusters in the analysis. Accessions belonging to papeda, *Fortunella*, and *P. trifoliata* are also in separate clusters. Interestingly, *C. hongheensis* CRC 3797, which is often classified in the papeda group, did not cluster with the other papeda accessions; instead, it nested with the pummelo cluster.

## 3. Discussion

High-density SNP genotyping arrays are a powerful method for characterizing elite plant germplasm and breeding populations [29,37,38,39]. The SNP array technology can provide adequate throughput and efficiency, issues which have limited previous studies of highly heterozygous organisms such as *Citrus*. SNP array technology takes advantage of SNPs, which are abundant, distributed across the genome, co-dominant, and likely to have high identity by descent rate. Therefore, high-density SNP arrays are useful not only to construct high-density genetic maps [40,41] but also for phylogenetic studies [42], association studies [43,44], and diversity studies [45] in a fairly high-throughput manner. For citrus, several arrays, including a 33,879 transcript assay Affymetrix™ GeneChip^®^ citrus genome microarray (Affymetrix™, Thermo Fisher Scientific, Santa Clara, CA, USA), 20K expression arrays [46], 1457 Goldengate SNP array [47], 384 marker SNP array [48], and 756 Goldengate SNP array [49], have been previously developed. These have been useful for many gene expression studies and for the detection of markers in phylogenetic research, genetic map construction, and trait mapping. In the genomic era, high-throughput genetic marker systems are essential for high-resolution genomic studies, and combined with available bioinformatics tools, they allow us to answer both biological and practical questions. Axiom^®^ Citrus HD Genotyping Array (3.9 SNP/kb) and Axiom^®^ Citrus Genotyping Array (0.16 SNP/kb) are the highest-density SNP genotyping arrays constructed and available for citrus so far. Further, these high-density SNP arrays allowed us to compile an extensive genomic inventory of the accessions in the CVC in a very high-throughput manner and perform a range of high-resolution genomic studies. As discussed previously, the heterozygous *Citrus* genome makes the GBS approach computationally demanding and, depending on sequencing cost, often more expensive per sample than in inbred plants. Alternatively, high-density SNP genotyping arrays will not only provide the coverage many genomic analyses require but are also efficient and have a high throughput. After the initial investment, the time required to go from sample to data is relatively short.

Although fixed array platforms such as those reported here have a wide variety of applications, they have limitations that can be addressed by other tools. A SNP array is limited to the detection of SNPs previously discovered by the diversity panel. Two citrus species not represented in our panel are *C. ichangensis* and *C. ryukyuensis* [50], the latter not yet discovered when the array was designed. Analysis of genotypes derived from these species may be inaccurate or lack sufficient depth for some applications. This problem is not easily addressed, although the magnitude of the error could be determined by comparing a sequence-based phylogeny of the ancestral taxa with that from array data by extracting SNP calls from the sequence for the array content. For linkage mapping and GWAS, the number of polymorphic or heterozygous SNPs in relatives such as *Poncirus* or *Microcitrus* may limit the power of the analysis. For these situations, resequencing should be useful and is now becoming cost-effective compared with arrays. Genotyping-by-sequencing and targeted resequencing are additional alternatives. If only modest (10 to 200) numbers of SNPs are required to address a question, then various alternative SNP genotyping methods may be suitable, such as KASP or high-resolution melt (HRM). The set of SNPs tested with these arrays can serve as a preselected set of single-copy sequences with documented polymorphisms from which to choose smaller sets of SNPs for other marker systems. For example, if a high density of SNPs polymorphic in a specific region of the genome is needed, these can be selected from those in the PHR class on the high-density SNP array, and SNP calls from this array may be useful to predict whether the SNP is likely to be polymorphic in the target genotypes.

### 3.1. Accuracy of Array Data in Citrus and Citrus Relatives

Our study also showed that high-density SNP genotyping can produce data and results that are comparable to those of whole genome sequencing for inferring all but fine-scale relationships while providing sufficient genome coverage with heterozygous markers for mapping in many citrus populations (Figure 2). SNPs represented on the array also targeted mostly genic regions (Axiom^®^ Citrus HD Genotyping Array: 49%, Axiom^®^ Citrus Genotyping Array: 66%) (Figure 2, Appendix A) in order to increase the success of SNP genotyping in diverse accessions of citrus. Due to the lower concordance observed in more distant taxa between array data and sequence data, a simulation study was conducted to show the effect of genotyping errors in *Poncirus* and its hybrid with *Citrus* within the range observed in array data. The simulation study showed that genotyping error ranging from 0 to 5% had little to no effect on the results of PCA (Appendix A), phylogeny (Appendix A), and Admixture analysis (Appendix A). Overall, data obtained from our study indicates that these arrays are most effective when applied to *Citrus* rather than its close relatives, probably due to the use of sequence from Clementine in probe design.

### 3.2. Population Structure

The population must be carefully examined to assess the appropriate marker density and to avoid spurious associations prior to performing association studies. PCA provides a multivariate tool [51] to analyze complex genomic variation data and estimate genomic divergence between populations. A large SNP dataset can be easily analyzed by PCA. Many accessions in the CVC have never been characterized with high-density marker systems prior to our study. PCA allows us to efficiently determine the genomic background of such taxa. The smaller, more economical array Axiom^®^ Citrus Genotyping Array provided sufficient marker density to perform PCA and successfully separated ancestral *Citrus* [9,35] and more distant taxa into separate clusters. *C. reticulata*, *C. maxima*, *C. medica*, and *P. trifoliata*, a *Citrus* relative often used in rootstock breeding, clustered separately, explaining the most variation within the dataset (59.6%) (Figure 7). PC1 mainly separated *C. medica* from the other accessions, PC2 mainly separated *C. maxima* from the rest of the dataset, and PC3 mainly separated *P. trifoliata* from *Citrus.* Percentage variation explained by PC3 in the Axiom^®^ Citrus HD Genotyping Array (4.1%) and the Axiom^®^ Citrus Genotyping Array (11%) differed significantly (Figure 6 and Figure 7), probably resulting from the inclusion of a greater proportion of *Poncirus* and *Poncirus* hybrid accessions in the Axiom^®^ Citrus Genotyping Array data (4.6% for the Axiom^®^ Citrus HD Genotyping Array data and 7.5% for the Axiom^®^ Citrus Genotyping Array data). In addition, our study with the Axiom^®^ Citrus Genotyping Array found that some accessions included in the study that were previously categorized as non-hybrid accessions, according to the classification currently employed in the CVC, are instead hybrids with other *Citrus*. Many cultivated accessions of *C. reticulata* have been reported to be introgressed with *C. maxima* [7,9]. PCA shows that the accessions identified as not introgressed in previous studies [9,32,33] clustered at the tip of the *C. reticulata* clusters. Preliminary analysis with Admixture and percent heterozygosity confirmed these results (data not shown). Reclassification of many accessions in the CVC with unknown genomic backgrounds would have been a laborious task had they and a large, diverse collection not been analyzed with such high-throughput SNP genotyping arrays, suggesting the great potential utility of the Axiom^®^ Citrus Genotyping Array.

### 3.3. Inference of Citrus Phylogeny with Array Data

Resolving *Citrus* lineage and phylogeny has long been an interest of citrus breeders and researchers. Phylogenetic studies are useful both in terms of genetic diversity analysis and parental line selection in breeding programs. Historically, the phylogeny and taxonomy of *Citrus* have been studied based on morphology and geography. Until recently, there have been two widely accepted *Citrus* taxonomic systems: Swingle and Reece; Tanaka. Comparing these two classical *Citrus* classifications, there are 16 species in two subgenera (*Citrus* and *Papeda*) proposed by Swingle [6] and up to 162 species proposed by Tanaka [52]. The major discrepancy between the two classical systems exists mainly in *Citrus* and stems from how much divergence within a recognized species should be allowed and, more specifically, whether natural hybrids should be considered as true species. Over the years, various molecular markers such as simple sequence repeats (SSR), restriction fragment length polymorphism (RFLP), random amplified polymorphic DNA (RAPD), sequence characterized amplified region (SCAR), sequence-related amplified polymorphism (SRAP), mitochondrial DNA, chloroplast DNA, chromosomal variability, and nuclear gene sequences have been used in phylogenetic investigations into *Citrus* to try to better understand the origin and relationships of diverse accessions [15,53,54,55,56,57,58]. Analysis of markers [54,59] and sequencing indicate four major ancestral *Citrus* groups: citron (*C. medica*), pummelo (*C. maxima*), mandarin (*C. reticulata*), and papeda (*C. micrantha*) [9], while Scora and Barrett and Rhodes [34,35] only recognized the first three groups as ancestors of cultivated citrus. Conventional phylogenetic trees are used to represent the evolutionary history of taxa; however, complex evolutionary scenarios that involve hybridization, horizontal gene transfer, recombination, and gene duplication and loss violate the assumptions of these models. More than a thousand years of domestication led to hybridization and introgression of ancestral progenitors into the cultivated *Citrus* varieties. Therefore, the identification of appropriate accessions prior to such analyses is very important. The phylogeny analysis performed with 26 CVC accessions included known hybrids and admixed accessions. For example, the placement of Schaub rough lemon CRC 3879 differs in the phylogenetic trees generated with array data and sequence data. Rough lemon is thought to be a natural hybrid of a citron and a mandarin [34]. The Structure analysis of rough lemon with 24 SSR loci (275 alleles) showed that it derived a majority of alleles from citron and the rest from mandarin [15], but analysis with 123 SSR, SNP, indel, and cytoplasmic markers indicated equal contributions from citron and mandarin [60]. Schaub rough lemon is nested with orange and mandarins with array data, while it is nested with lemon, lime, and citron with sequence data. An F1 hybrid may cluster with either parent group so that sampling error can contribute to variation in its placement. There is 97.8% genotyping call concordance between Axiom^®^ Citrus HD Genotyping Array data and sequence data, and much of the discrepancy between these two trees, most notably the placement of Schaub rough lemon, is likely due to the inclusion of such hybrid accessions in the analysis. Nonetheless, the topological comparison showed that the Axiom^®^ Citrus Genotyping Array could provide data and information content comparable to that of sequence data in phylogenetic studies (Figure 8, Appendix A).

More definitive and informative evidence of ancestry can be provided by the determination of chromosome-length haplotypes of hybrid accessions combined with the identification and analysis of the chromosome distribution of species-specific SNPs.

### 3.4. Inference of Phylogeny of 87 Non-Admixed Accessions of Citrus and Close Relatives

Previous studies of *Citrus* phylogeny often resulted in an insufficient understanding of the genetic diversity that exists in *Citrus* species due to inadequate sampling of markers or accessions. The CVC provides an excellent opportunity to assess genetic diversity and population structure, given its size and diversity. In addition, the use of whole genome SNP markers provided by Axiom^®^ citrus arrays should significantly improve the resolution of phylogenetic study in *Citrus*, given their genome coverage and SNP targets. While rapid progress has been made in this field in many other plants and some crops, progress in *Citrus* has mostly been limited to the major commercially important groups that have been characterized by whole genome sequencing [9]. Our phylogenetic analysis using Axiom^®^ Citrus array data of 87 non-admixed accessions in the CVC is the most extensive phylogenetic study of non-admixed *Citrus* accessions to date [9,54,57,59,61,62,63,64,65]. Phylogeny inferred with Axiom^®^ Citrus array data identified five monophyletic clusters representing citron, pummelo, mandarin, kumquat, and papeda groups with 100% bootstrap support (Figure 9), consistent with previous reports that are based on fewer markers or accessions [9,15,58,66]. *C. tachibana* CRC 3150 is in one of the minor subclusters and most different from other mandarin accessions, as inferred previously [9,61,62]. A study with SSRs reported *C. tachibana* clustered with papeda accessions but without strong bootstrap support [15]. Increased marker density was likely able to better resolve the relationships of *C. tachibana* in the mandarin cluster. A recent analysis of whole genome sequence data [9] concluded that *C. tachibana* should be designated a subspecies of *C. reticulata.* Yellow rind mandarin CRC 3895 is in another minor subcluster and is also a rather divergent mandarin type. There has been no report of marker studies performed on yellow mandarin; however, it is thought to be an older mandarin from China, according to the description on the USDA Germplasm Resources Information Network (https://www.ars-grin.gov/ (accessed on 27 December 2023)). All accessions of *Fortunella* and the papeda group analyzed in the phylogeny study form two clusters with very short branch lengths between accessions; hence, these accessions are similar to one another, except for *C. hongheensis* CRC 3797. *C. hongheensis* CRC 3797 has been considered to be a papeda but clustered as a sister group to pummelos. A previous study with RFLP and RAPD showed that *C. hongheensis* CRC 3797 clustered with pummelo [53]. Its heterozygosity is 2.4% and lower in comparison to *C. hystrix* CRC 3103 and *C. micrantha* CRC 3605, which are in the citrus variant discovery panel. The average heterozygosity of pummelo with Axiom^®^ Citrus 56AX genotyping array is 8.2%. Sequence analysis of selected genic regions of chloroplast DNA also clustered papedas such as *C. latipes* and *C. ichangensis* with pummelo [67]. Papeda is one of the two subgenera within *Citrus* defined by Swingle [6], but other studies suggest that papeda is a polyphyletic group [58,68]. Although *C. hongheensis* CRC 3797 clusters with pummelo, it is very divergent, indicating the possibility that *C. hongheensis* CRC 3797 could be a sister species to pummelo. However, it is also possible that the low heterozygosity observed is due to ascertainment bias introduced by the citrus variant discovery panel not including a divergent papeda, *C. ichangensis*. *C. ichangensis* belongs to subgenus *Papeda* by Swingle’s classification system [6] based on the presence of acidic oil in the fruits and morphology; however, Tanaka placed *C. ichangensis* together with all mandarin species [5]. *C. ichangensis* is thought to be a more genetically divergent type in the papeda group [52]. The possible chloroplast genome relationship between *C. ichangensis* and *C. reticulata* is reported by marker analysis of chloroplast DNA [54]. More recently, whole genome sequencing analysis revealed *C. ichangensis* to be an ancestral species [50]. Our phylogeny analysis conducted with array data including *C. ichangensis* and 500 chloroplast SNP array data also indicates *C. ichangensis* is related to *C. reticulata* rather than to the papeda group or *C. hongheensis* (data not shown). Therefore, we suspect that the low heterozygosity of *C. hongheensis* could be an artifact resulting from ascertainment bias. It is possible that *C. hongheensis* is a hybrid between a pummelo and a papeda type that is not represented in the citrus variant discovery panel. The array data show that both citron and pummelo clusters have little within-group population structure. Our phylogeny analyses strongly suggest that the array data could provide resolution and information content comparable to those of sequence data in phylogenetic analyses with smaller sample sizes (Figure 9, Figure 10 and Appendix A). Both phylogeny trees generated with array data and sequence data cluster the parent–offspring pair Siamese Acidless pummelo CRC 2240 and Chandler pummelo CRC 3224 together. In addition, in both trees, Yunnanese citron, *C. tachibana* mandarin, and Hunan pummelo are the most divergent accessions in their species’ clusters. Phylogenetic trees constructed in our study with Axiom^®^ citrus array data and previously reported trees often differed in relative branch lengths between the identified six clusters for several reasons. First, the SNPs chosen from the citrus variant discovery panel were all polymorphic and mainly designed to work well in *Citrus*; therefore, an overestimation of divergence between different *Citrus* clusters could occur. Secondly, the array was designed to discard the rarest alleles and relatively few accessions of relatives, and some groups, such as papedas, were included, perhaps leading to underestimates of divergence between these taxa and other citrus. Lastly, we performed limited LD filtering in the analyses to reduce the bias due to redundant variants. Higher LD levels can lead to overestimation of distances between clusters. LD can be created as a consequence of mutation, genetic drift, selection, and limited recombination, which lead to population structure due to differences in allele frequencies between groups when individuals from different genetic origins are included in the dataset [69,70]. Although the extent of LD has not been thoroughly investigated in *Citrus* with high-density marker systems, it is presumed to vary greatly depending on the population [10]. For *Citrus* species with self-incompatibility, LD is likely to be shorter than LD in the population of interspecific hybrids and admixed accessions [71]. A report of LD in citrus based on 1841 SNPs showed a similar trend [10]. Therefore, SNP density represented on the Axiom^®^ Citrus HD Genotyping Array and the Axiom^®^ Citrus Genotyping Array should be sufficient for downstream applications such as linkage mapping, ancestry analysis, and, perhaps, genomic selection, that do not require very high marker densities.

## 4. Materials and Methods

### 4.1. Sample Preparation for Array Hybridization

Leaf samples (young flush) of the accessions of *Citrus* and its relatives (Appendix A) were collected on ice in the field from the Citrus Variety Collection (CVC, Riverside, CA, USA) from 2014 to 2016. Each tree from which a leaf sample was collected was tagged with a unique identifier upon collection. Collected leaf samples were then returned to the laboratory on ice, and the leaf surface was quickly and gently wiped with chloroform prior to drying with silica packs in a sealed, labeled sample bag. Silica-dried leaf samples were homogenized with FastPrep-24™ (MP Biomedicals, Santa Ana, CA, USA) with ¼” ceramic spheres or ¼” stainless steel beads. High molecular weight genomic DNA was then isolated from homogenized dried leaf samples with DNeasy^®^ Plant Mini Kit (Qiagen, Hilden, Germany). The quality of DNA was assessed by electrophoresis, 260/280 and 260/230 ratios with a NanoDrop spectrophotometer (Thermo Fisher Scientific, Wilmington, NC, USA), and genomic DNA was quantified with a Qubit fluorometric system (Thermo Fisher Scientific, Wilmington, NC, USA) according to manufacturer’s instructions. Genomic DNA of concentration of 25 ng/µL in 50 µL and 30 ng/µL in 25 µL for Axiom^®^ Citrus HD Genotyping Array and Axiom^®^ Citrus Genotyping Array, respectively, were prepared in Affymetrix™ bar-coded 96 well plates. Samples were submitted to Affymetrix (Santa Clara, CA, USA, now part of ThermoFisher) to be amplified, fragmented, and hybridized on the arrays, followed by single-base extension through DNA ligation and signal amplification, following the standard protocol for the Affymetrix Axiom^®^ Array. The Affymetrix GeneTitan^®^ platform was used for genotyping the “Reference Set” with the SNP array.

Design criteria for Citrus SNP arrays included (1) the ability to call SNPs accurately in the diverse accessions of Citrus and close relatives that sexually hybridize with Citrus, (2) sufficient nuclear genome coverage, and (3) inclusion of chloroplast SNP markers. SNP marker development for genotyping arrays involves the following: (1) selection of a diverse set of genotypes to be included in the citrus variant discovery panel, (2) variant discovery using resequencing data, (3) variant validation, and (4) final selection of variants to be represented on the array (Figure 10). Variant validation used the Axiom^®^ Citrus HD Genotyping Array to select a subset of robust SNPs to represent on the Axiom^®^ Citrus Genotyping Array.

### 4.2. Variant Selection According to In Silico Analysis of Re-Sequencing Data

The citrus variant discovery panel was composed of 30 accessions (Table 1) that were whole genome re-sequenced with Illumina^®^ HighSeq 2500 at the UCR Institute for Integrative Genome Biology (IIGB, Riverside, CA, USA) genomics core, as well as whole genome sequence data of 11 accessions taken from publicly available databases and from a cooperating laboratory (National Institute of Genetics, Shizuoka, Japan) (Table 2). For whole genome sequencing, young leaf tissue was sampled from trees growing in the UCR Citrus Variety Collection, now called Givaudan Citrus Variety Collection (Riverside, CA, USA), and DNA was isolated from tissue ground with liquid nitrogen in a mortar and pestle using Qiagen DNeasy^®^ Plant Mini Kit (Qiagen, Hilden, Germany). Sequencing libraries were prepared at IIGB using the NEBNext Ultra DNA library prep kit for Illumina (NEB) and sequenced (100 bp paired-end reads) with a target depth of at least 30×. Variant sites were mined using a variant discovery pipeline described below at the DOE Joint Genome Institute (JGI) using Phytozome *C. clementina* v1.0 reference genome http://www.phytozome.org/citrus/ (accessed on 15 April 2015) for nuclear genome variant discovery and by mapping reads to the sweet orange chloroplast genome [72] for chloroplast variant discovery. We considered this the best assembled and annotated genome available at that time, and the UCR germplasm collection contains more mandarins than any other non-clonal group. The Illumina^®^ paired-end reads were mapped to the haploid clementine reference sequence using bwa/mem, followed by the removal of duplicated reads using Picard Mark Duplicates version 1.92. The Genome Analysis Toolkit (GATK) HaplotypeCaller version 3.3-0-g37228af was used for SNP calls with the following filters: map quality score ≥ 25, base quality score ≥ 30; sample genotype quality score ≥ 20; read depth filter for each sample: 10 ≤ DP < 2× median; allele balance filter for heterozygous SNPs: alt allele frequency from reads of a sample does not reside in the tails of a binomial distribution accounting for 5% of total area. The final variant set was restricted to sites with no more than 2 alleles and fewer than half of the samples having missing genotypes [51].

### 4.3. Variant Filtering Part 1

Variant Call Format (VCF) files of 41 accessions in the citrus variant discovery panel were generated from the Illumina^®^ short read data and grouped into Group 1 (Pummelos, mandarins and some citrus hybrids, 29 accessions), Group 2 (Citron, 4 accessions), and Group 3 (Citrus relatives, 8 accessions, including 2 papedas) based on the sequence divergence between groups, calculated prior to SNP filtering (data not shown). SNPs were assigned to to the phylogenetic group or groups in which they were polymorphic (Alt allele frequency > 0.1) to allow us to select a more appropriate number of SNPs for analysis of each group. Without some limitation of this type, most SNPs on the array might be useful for differentiating the divergent groups but not useful for analysis within the core group of *Citrus* accessions in which most breeding is performed. After reviewing the group analysis we concluded that the distribution of SNPs within and among groups was acceptable without additional filtering by group. SNP filtering was then performed using SnpEff snpeff.sourceforge.net (accessed on 28 August 2015) and SnpSift snpeff.sourceforge.net/SnpSift.htm (accessed on 28 August 2015), mainly targeting putative genic regions, which included exons, introns, 5′ and 3′ UTRs, 5 kb upstream, and downstream regions of a putative gene. This strategy was expected to select variants more likely to affect gene functions and have flanking sequences relatively conserved among the target taxa to reduce miscalls due to multiple SNPs. A total of 6,931,772 autosomal variants resulted from further SNP filtering based on the alternate allele frequency (>0.1) in Groups 1, 2, or 3. Next, A/T or G/C transversion variants were removed from these filtered autosomal variants as recommended by the Affymetrix™ bioinformatics team, resulting in 5,320,680 autosomal variants, including 639,833 InDels (Appendix A). Chloroplast variants were not filtered at this stage, and all were added to the autosomal variants. The filtered candidate variants were submitted to the Affymetrix™ bioinformatics team for in silico analysis (Appendix A) to predict variant to probe conversion performance for the array.

#### 4.3.1. In Silico Analysis of Variants

Upon Affymetrix™ in silico analysis of 5.3M (5,320,680 autosomal and 2215 chloroplast) candidate variants, 436,431 variants were recommended for tiling on both strands, 1,599,832 variants for tiling only on one strand, 3,285,540 neutral or not recommended variants that could be tiled on at least one strand, and 1092 variants that could not be tiled because the probes for these loci could not be created. Overall, 2,036,263 variants (38.25%) were recommended for tiling on at least one strand. The 639,833 InDels failed to pass Affymetrix™ in silico analysis and were excluded from further filtering at this stage. In order to perform further filtering of the candidate variants with maximal efficiency, we focused on filtering from the 2,036,263 variants that were recommended for tiling on at least one strand. Further filtering was performed based on the Pconvert value, an Affymetrix metric that predicts whether a polymorphism can be genotyped accurately. If both forward and reverse strand tiling probes had the same Pconvert value, the variants were accepted to be tiled on both strands. If both forward and reverse strand tiling probes were recommended but Pconvert values differed, the strand with the higher Pconvert value was selected if this value was above 0.818. Finally, Pconvert values above 0.69 were used to accept the rest of the variants to be tiled on either the forward or reverse strand or both (for 106,707 variants). No further filtering was performed on chloroplast variants, and 1002 Affymetrix™ recommended chloroplast variants were accepted based on the in silico analysis. Affymetrix™ Axiom^®^, genotyping array technology, uses probes of 71 bases to query the targeted SNP, and probes on the Axiom^®^ Citrus HD Genotyping Array were designed to match the *C. clementina* v1.0 reference genome sequence except at the targeted SNP position base.

#### 4.3.2. SNP Validation Panel

To assess whether the selection of a robust set of variants that work in both *Citrus* and its near relatives was achieved, a total of 277 genomic DNA samples were selected from CVC as a validation panel and screened with Axiom^®^ Citrus HD Genotyping Array. Samples from 3 breeding parents, 4 duplicates, 8 polyploids, and 8 synthetic polyploids (mixed DNA) were removed from the set of 277 genomic DNA samples for this additional analysis because the Axiom^®^ Analysis Suite software version 3.1.51.0 calls SNPs only in diploids or haploids, and genotyping breeding parents was a low-priority objective. Triploid and tetraploid forms rarely occur in citrus, and the existing Axiom software is not designed to accurately call SNPs in which allele dosage in heterozygotes is not 1:1, although custom scripts have been developed for allo-octoploid strawberry [27]. The remaining 254 samples (Appendix A) were then analyzed. This process allowed performance assessment of the represented SNPs and the array but was also essential for identifying additional potential SNP filtering parameters. In addition, we assessed the range of phylogenetic divergence over which the selected SNP probes were reliable. Trios (7 families) (Appendix A), technical, clonal duplicate and polyploid, and synthetic polyploid (mixed DNA) samples were included in the validation panel. SNPs and array performance were investigated with respect to (1) SNP genotyping call rates, cluster separation, and reproducibility; (2) level of heterozygosity in the validation panel; (3) and mendelian inheritance from parents to offspring in trios (Appendix A). 

### 4.4. Variant Filtering Part 2

To design an array that accurately genotypes SNPs in both *Citrus* and its close relatives, we performed variant filtering part 2. In analysis 1, SNP genotyping calls for 254 genomic DNA samples analyzed with Axiom^®^ Citrus HD Genotyping Array were made using Axiom^®^ Analysis Suite 1.1.1.66 by Affymetrix™ at Axiom^®^ BestPractice workflow default setting (default DQC, QC, and % passing) to identify “*PolyHighResolution*” (PHR) probes. PHR probes are the most informative and accurate probe performance class on the Axiom^®^ platform based on the genotype call cluster separation, cluster variance, and cluster position.In analysis 2, SNP genotyping calls of 254 diploid genomic DNA samples were made for PHR probe loci identified in analysis 1 using Axiom^®^ Analysis Suite at lowered DQC, QC, and % passing thresholds (=0) to allow Axiom^®^ Analysis Suite to analyze all the diploid samples including citrus relatives which otherwise failed the analysis due to their divergence from the majority of the samples being analyzed. DQC and QC probes are used by the Affymetrix™ Axiom^®^ platform as quality controls. DQC probes are 31-mer sequences that are non-polymorphic in the variant discovery panel, and QC probes are the best-performing probes selected by Affymetrix™ in silico analysis. Among the 20,000 QC probes, the number of probes polymorphic in various groups were 15,539 in Group 1 only, 786 in Group 2 only, 120 in Group 3 only, 1181 in Groups 1 and 2, 487 in Groups 1 and 3, 78 in Groups 2 and 3, and 1809 polymorphic in all three groups (Appendix A).R package SNPolisher (SNPolisher, v1.5.2, Affymetrix™) was used to perform the “BalleleFreq test”, which is a post-processing function to check if there is a shift in intensity in probes across samples. When samples are processed at the same time, an intensity shift in a small number of probes in some of the samples can, in fact, cause a sample to be assigned to the wrong cluster. BalleleFreq test is performed by comparing the SNP genotyping calls between analysis 1 and analysis 2 at the PHR probe loci in each individual analyzed. We then selected loci for which calls did not differ between analysis 1 and analysis 2. Analysis 1 identified 728,831 PHR probe loci that accurately genotype SNPs in Citrus and analysis 2 identified 369,269 PHR probe loci that accurately genotype SNPs in both *Citrus* and its relatives.

In order to select high-confidence variants to be represented on the smaller and more economical Axiom^®^ Citrus Genotyping Array, we selected the best-performing probes from 369,269 PHR probe loci identified in variant filtering parts 1 and 2 based on (1) the best-performing probe from each pair for which probes were tiled on both strands and the (2) concordance between in silico variant calls from the sequencing data of the variant discovery panel and SNP genotyping calls with validation array Axiom^®^ Citrus HD Genotyping Array. We then randomly selected 58,433 variants from the list and assessed the distribution of the selected 58,433 SNPs along the genome and the number of heterozygous variants in selected genotypes. Finally, a total of 58,433 SNPs (57,933 autosomal SNPs and 500 chloroplast SNPs) were chosen to be represented on the Axiom^®^ Citrus Genotyping Array.

### 4.5. Heterozygosity and Reproducibility of Genotyping Data

The reproducibility of the data generated by the arrays was assessed by using heterozygosity and concordance of genotyping data generated by Axiom^®^ Citrus HD Genotyping Array and Axiom^®^ Citrus Genotyping Array. Percent average heterozygosity among polymorphic sites in the sequence data was estimated from the 19M VCF file and compared to that obtained for the same accessions with the Axiom^®^ Citrus HD Genotyping Array. In addition, concordance of genotyping calls between the sequence data and Axiom^®^ Citrus HD Genotyping Array data and between Axiom^®^ Citrus HD Genotyping Array data and Axiom^®^ Citrus Genotyping Array data was calculated to assess the reproducibility of our data. The loci where genotyping calls were not made, either by both data sets or by one data set, were omitted from these calculations.

### 4.6. PCA

PCA was carried out with both Axiom^®^ Citrus HD Genotyping Array and Axiom^®^ Citrus Genotyping Array genotyping data. PCA with the Citrus HD Genotyping Array analyzed genotyping calls at 728,831 PHR loci of 196 diverse genomic DNA samples after eliminating duplicate samples used to test reproducibility and those samples that had QC at <97% (Table 5 and Appendix A). PCA was also carried out with genotyping calls at 51,296 PHR loci for combined data of both arrays from 871 CVC accessions (Table 5 and Appendix A). The PCA analyses were carried out with the software TASSEL v5.2.31 http://www.maizegenetics.net/tassel (accessed on 21 April 2016).

### 4.7. Phylogenetic Inference

Four phylogenetic analyses of the citrus variant discovery panel and CVC accessions representing major *Citrus* taxa and close relatives were performed. (1) Genotyping calls of the Axiom^®^ Citrus HD Genotyping Array and the Axiom^®^ Citrus Genotyping Array and sequence data of 26 selected accessions (Appendix A) were used to infer phylogenetic relationships among accessions in the citrus variant discovery panel. The 26 accessions were selected based on samples passing at the default setting with the Axiom^®^ Analysis Suite and no more than 20% missing calls with the sequence data. Genotyping calls of the Axiom^®^ Citrus HD Genotyping Array and the Axiom^®^ Citrus Genotyping Array were filtered for PHR. (2) In addition, PHR loci from the Axiom^®^ Citrus Genotyping Array were pulled from the sequence data and used for a phylogenetic analysis. The resulting phylogenetic trees generated with array data and sequence data were then compared to each other topologically with phylo.io https://phylo.io (accessed on 2 July 2017). (3) For phylogenetic analysis of non-hybrid and non-admixed or minimally admixed CVC accessions, 87 accessions representing major citrus taxa and close relatives were selected (Appendix A). The SNP genotyping dataset was filtered for PHR, and all 87 accessions that had less than 10 percent heterozygosity were selected (a threshold supported by preliminary Admixture analysis, not shown). A preliminary phylogenetic analysis using 871 CVC accessions grouped the 33 accessions belonging to the important *Citrus* relative, *P. trifoliata*, into three major clusters (Appendix A) with little genetic variation within each cluster (data not shown). Only one individual representing each of the three *Poncirus* clusters was included in the analysis to reduce redundancy. The accession of *C. micrantha* var. *microcarpa* failed the initial analysis with Axiom^®^ Analysis Suite when analyzed with default settings. Therefore, the QC call rate threshold was reduced from 97% to 95% in Axiom^®^ Analysis Suite to include *C. micrantha* var. *microcarpa* in the phylogenetic analysis. The pipeline SNPhylo [73] was used to perform phylogenetic analysis with *Poncirus* as an outgroup genus. SNPhylo first concatenated and aligned the SNPs, then performed a maximum likelihood distance estimation without a molecular clock and bootstrapping analysis [73]. For SNPhylo, the parameter settings were as follows: the minor allele frequency filter was set to 0, the maximum missing data filter was set to 1, and the LD filter was set to 1 to remove redundant SNPs by SNPrelate [74]. The maximum likelihood tree by DNAML [75] with bootstrap analysis with 1000 replications by Phangorn [76] was constructed. The phylogenetic tree was then generated and visualized using FigTree v1.4.3. http://tree.bio.ed.ac.uk/software/figtree/ (accessed on 12 October 2017). (4) A neighbor-joining phylogeny tree of 26 selected accessions in the citrus variant discovery panel was generated with sequence data that are not filtered for Axiom array loci. SNPhylo was used to filter 2,251,517 SNPs from the 19M SNPs in the full SNP database. For parameters for SNPhylo, the minor allele frequency filter was set to 0, the maximum missing data filter was set to 1, and the LD filter was set to 1. In addition, quality filtering of 5 and 5 for maximum PLCS (the percent of low coverage sample) and minimum depth coverage, respectively, was applied. A neighbor-joining tree was then generated with TASSEL v5.2.31 and visualized with Dendroscope 3 [77].

### 4.8. Simulation Study

A genotyping error simulation study with the Axiom^®^ Citrus Genotyping Array data on 26 selected CVC accessions (Appendix A), including 21 accessions in the citrus variant discovery panel and Carrizo Citrange 2863, was performed for PCA, phylogeny, and Admixture analyses to understand the effect of genotyping errors in selected analyses. Genotyping errors of 0, 3, and 5% were introduced in Flying Dragon trifoliate 3330A and Carrizo citrange CRC 2863 because these accessions have a higher occurrence of genotyping errors. Carrizo citrange 2863 was not represented in the discovery panel; therefore, haplotype data inferred from whole genome amplified single pollen grains genotyped with the Axiom^®^ Citrus Genotyping Array (data not shown) was used as a substitute for the sequence data. Genotyping data with 0% error were generated by excluding all missing calls and calls not identical between the array and sequence calls, resulting in a total of 14,470 loci used in the simulation study. Synthetic errors were then introduced by selecting random 3 and 5% of loci and changing heterozygous calls to homozygous and homozygous calls to heterozygous. PCA was conducted following the method described above. Phylogeny was inferred, and a neighbor-joining tree was generated using TASSEL v5.2.31 http://www.maizegenetics.net/tassel (accessed on 21 April 2015). Statistical comparison of trees was performed with R package APE [78] and pipeline ETE3 [79]. Admixture was performed using the software Admixture v1.3 with 10-fold cross-validation [80].

## 5. Conclusions

The high-density Affymetrix™ Axiom^®^ Citrus HD Genotyping Array and the Axiom^®^ Citrus Genotyping Array were developed with more than 1.4M and 58K SNPs that are well distributed over the citrus genome. Our study included many accessions in taxa relatively distant from *C. clementina*. Sequence read alignment for variant discovery in 41 accessions and probe sequence design for Axiom^®^ citrus arrays used the *C. clementina* v1.0 reference genome, and this is likely the cause for the lower concordance between sequence and array data in some taxa (Figure 4 and Figure 5). Lower concordance in the more distant taxon *Poncirus* was observed, and we suspect that one or more additional adjacent SNPs or indels in the genomes of more distant taxa interfered with hybridization. Nonetheless, our study shows that the Axiom^®^ arrays for *Citrus* and some related genera provide a tool that is very powerful and useful and increases throughput in analyzing the diversity of complex citrus genomes. They also provide us with a standardized set of markers that we can use repeatedly for mapping traits in various populations. The utility of these arrays in GWAS, QTL studies, and high-density linkage map construction in many crosses to dissect important traits such as fruit quality, nucellar embryony, and disease resistance is promising, given the high-density of robust SNPs represented on the arrays. Such results should be useful to further assist marker-assisted breeding. In addition, the arrays are used in the detection of recent admixture events, population structure analysis, LOH analysis, haplotype construction, and investigation into the parentage of CVC accessions. Although phylogenetic analysis using array data may not always yield the correct interpretation of evolutionary events due to the ascertainment bias, both arrays include high-density chloroplast SNPs, 1002 and 500 SNPs, respectively, and therefore, will help improve our understanding of phylogenetic relationships between notoriously confounded citrus accessions. As improved reference genomes for citrus become available, it should be possible to improve the annotation of these arrays.

## Figures and Tables

**Figure 1 plants-13-00691-f001:**
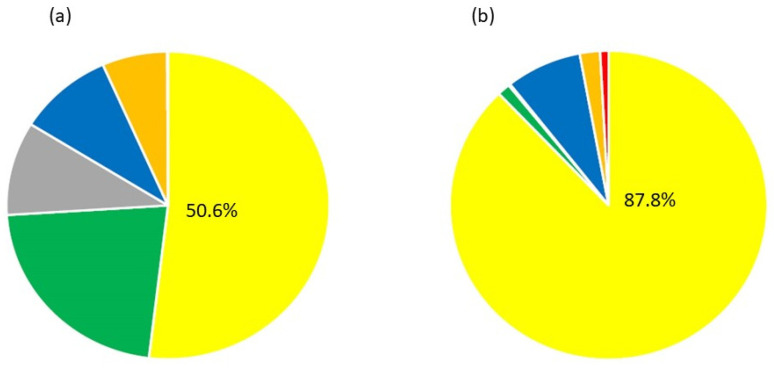
Proportion of markers on the Axiom^®^ Citrus HD Genotyping Array (**a**) and the Axiom^®^ Citrus Genotyping Array (**b**) classified in 6 categories by the Axiom Analysis Suite with default settings. Categories of markers are PolyHighResolution (yellow), OTV (Off Target Variant, gray), CallRateBelowThreshold (orange), Hemizygous (Red), other (green), and MonoHighResolution (blue).

**Figure 2 plants-13-00691-f002:**
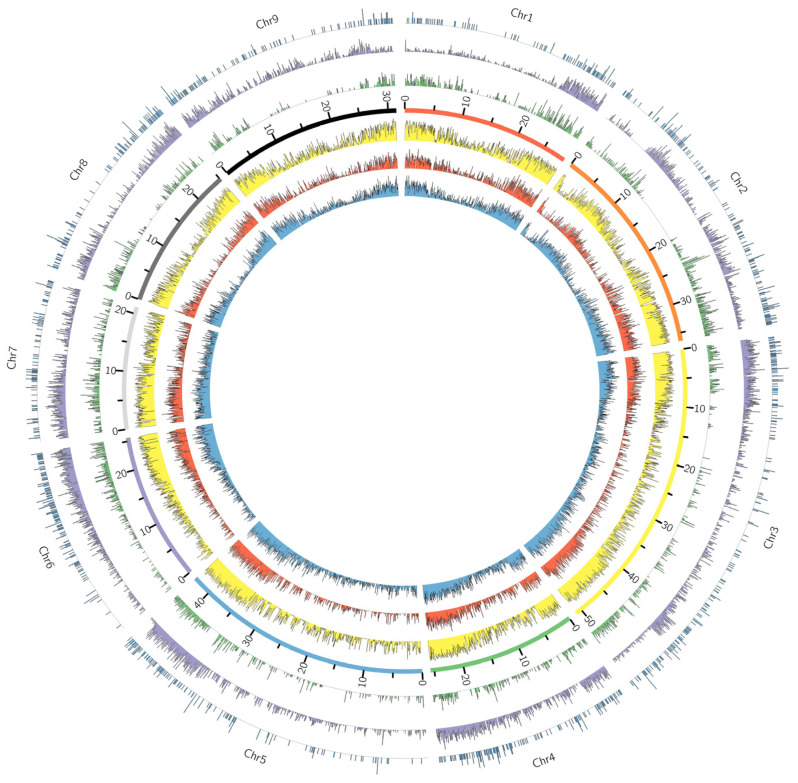
Genome-wide SNP distributions for the Axiom^®^ Citrus HD Genotyping Array, the Axiom^®^ Citrus Genotyping Array, and heterozygous marker distribution of selected accessions. Gene density of *C. clementina* was binned in 50 kb windows (blue), SNP density distribution of the Axiom^®^ Citrus Genotyping Array (red), and the Axiom^®^ Citrus HD Genotyping Array (yellow) on each of nine chromosomes from data binned in non-overlapping 50 kb windows. Distributions of heterozygous markers for Chandler pummelo (green), sweet orange (purple), and Pomeroy trifoliate (dark blue) on the Axiom^®^ Citrus Genotyping Array data were binned in 50 kb windows. There are 4554 heterozygous markers in Chandler pummelo, 18,159 heterozygous markers in sweet orange, and 1293 heterozygous markers in Pomeroy trifoliate.

**Figure 3 plants-13-00691-f003:**
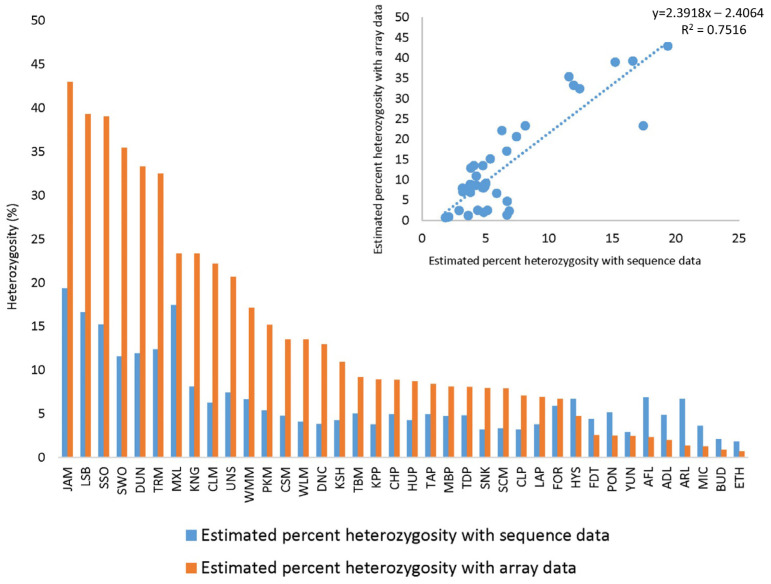
Comparison of percent heterozygosity estimated with sequence data and array data. Percent heterozygosity (proportion of 19 million variable sites called heterozygous in sequence data) was estimated for 38 accessions included in the citrus discovery panel (see Appendix A for abbreviations). The correlation between sequence data and array data for estimated percent heterozygosity is shown in the plot (upper right corner) with an R^2^ value of 0.7516.

**Figure 4 plants-13-00691-f004:**
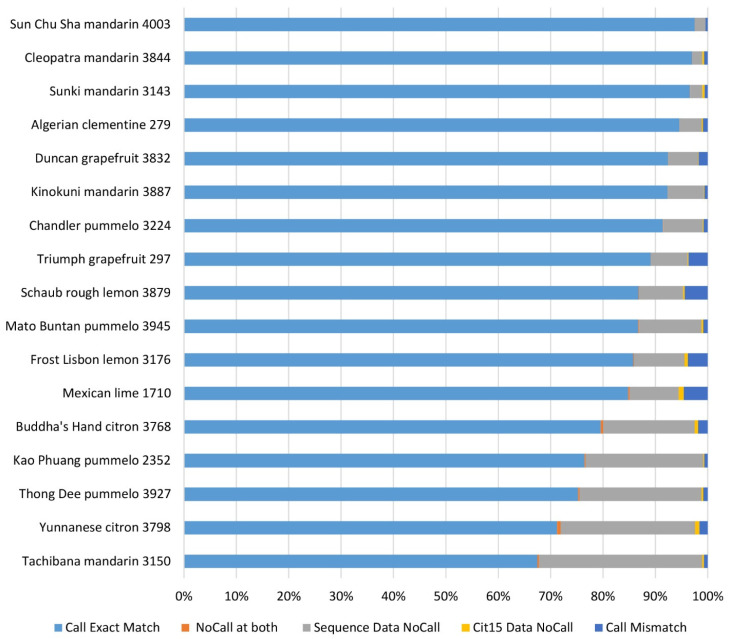
Genotyping call comparison between sequencing data and Axiom^®^ Citrus HD Genotyping Array data for 17 accessions from citrus variant discovery panel that were also genotyped with the Axiom^®^ Citrus HD Genotyping Array. Genotyping calls at 635,809 PHR loci were considered.

**Figure 5 plants-13-00691-f005:**
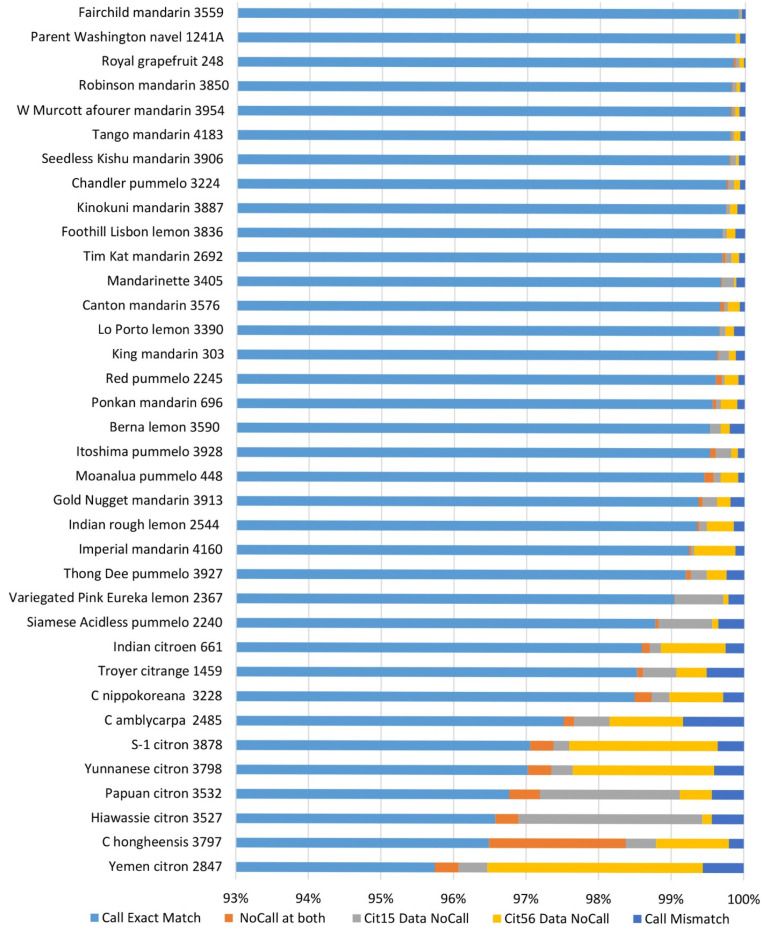
Reproducibility of genotyping data between the Axiom^®^ Citrus HD Genotyping Array and the Axiom^®^ Citrus Genotyping Array was assessed using 36 accessions that were analyzed with both arrays. Genotyping calls at 51,296 PHR loci were considered. Figure shows X-axis ranging from 93% to 100%.

**Figure 6 plants-13-00691-f006:**
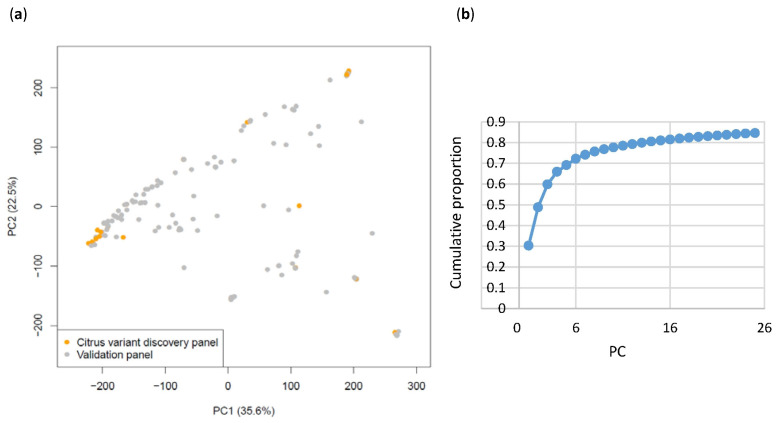
PCA of citrus variant discovery panel and accessions genotyped with the Axiom^®^ Citrus HD Genotyping Array. (**a**) PCA was carried out with 196 genomic DNA samples from the CVC (gray) genotyped at PHR loci with the Axiom^®^ Citrus HD Genotyping Array. Accessions representing the citrus variant discovery panel are highlighted in yellow. PC1 (35.6%), PC2 (22.5%), and PC3 (4.1%) explained 62.2% of the total variation that exists in the data. The individuals in the citrus variant discovery panel are shown (orange). (**b**) Cumulative proportion calculated for PCs.

**Figure 7 plants-13-00691-f007:**
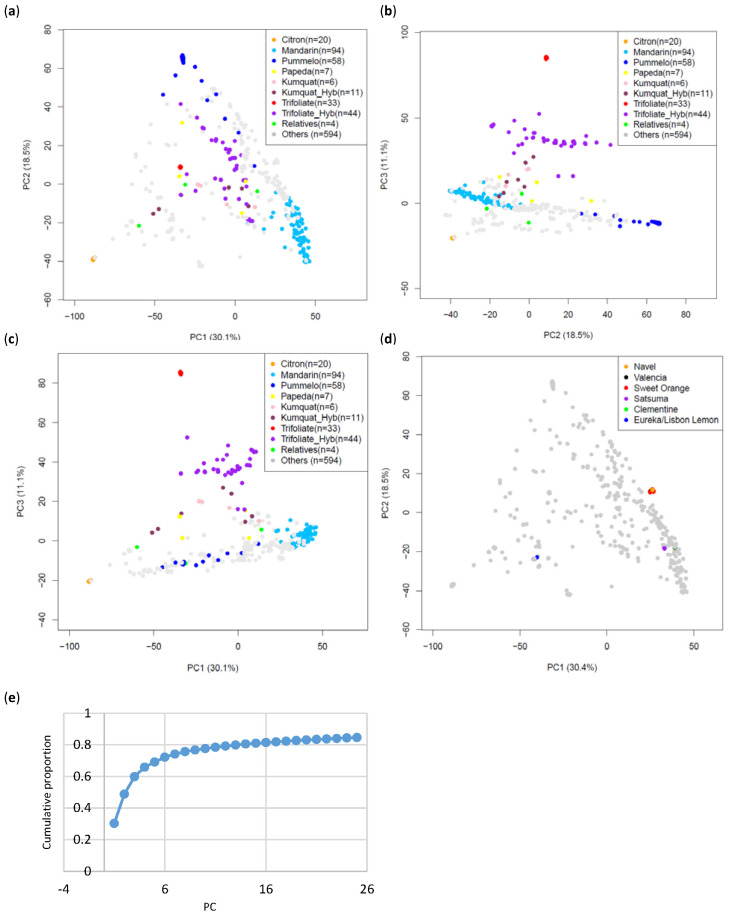
PCA of 871 diploid accessions in the CVC genotyped with the Axiom^®^ Citrus Genotyping Array at PHR loci. Accessions included 20 citrons (orange), 94 mandarins (light blue), 58 pummelos (blue), 7 papedas (yellow), 6 kumquats (pink), 11 kumquat hybrids (brown), 33 trifoliates (red), 44 trifoliate hybrids (purple), 4 citrus relatives (green), and 594 others (gray). PC1 (30.1%), PC2 (18.5%), and PC3 (11.1%) explained 59.7% of the total variation that exists in the data. (**a**) PCA PC1 vs. PC2. (**b**) PCA PC2 vs. PC3. (**c**) PCA PC1 vs. PC3. (**d**) PCA of clonally propagated accessions: navel oranges (orange), Valencia oranges (dark blue), sweet oranges (red), satsuma mandarins (purple), clementine mandarins (green), and Eureka/Lisbon lemons (blue). (**e**) Eigenvalues calculated for PCA.

**Figure 8 plants-13-00691-f008:**
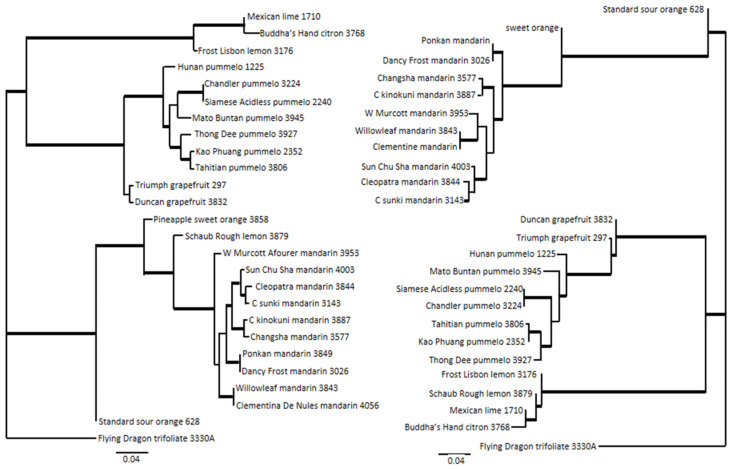
Maximum likelihood phylogeny trees of 26 accessions of *Citrus* and close relatives in the citrus variant discovery panel. Maximum likelihood trees were generated by pipeline SNPhylo and visualized by FigTree v1.4.3. Bootstrap values based on 1000 replications. Bootstrap values above 90% are indicated as bold lines. Tree generated by Axiom^®^ Citrus Genotyping Array data with 41,626 concatenated SNPs on the left and tree generated from sequence data with 32,754 SNPs that were also analyzed with Axiom^®^ Citrus Genotyping Array on the right.

**Figure 9 plants-13-00691-f009:**
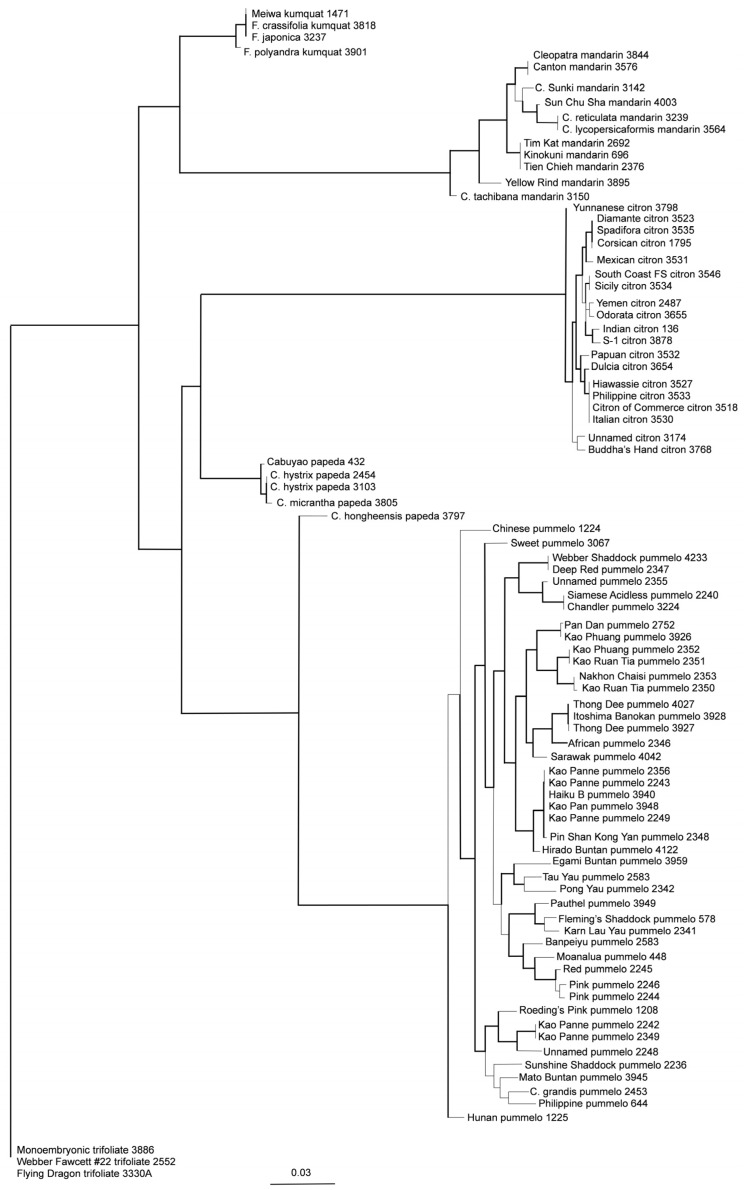
Maximum likelihood phylogeny tree of 87 non-admixed accessions in Citrus and close relatives with the Axiom^®^ Citrus Genotyping Array data on 41,626 concatenated SNPs. Bootstrap values of above 90% are indicated by bold lines.

**Figure 10 plants-13-00691-f010:**
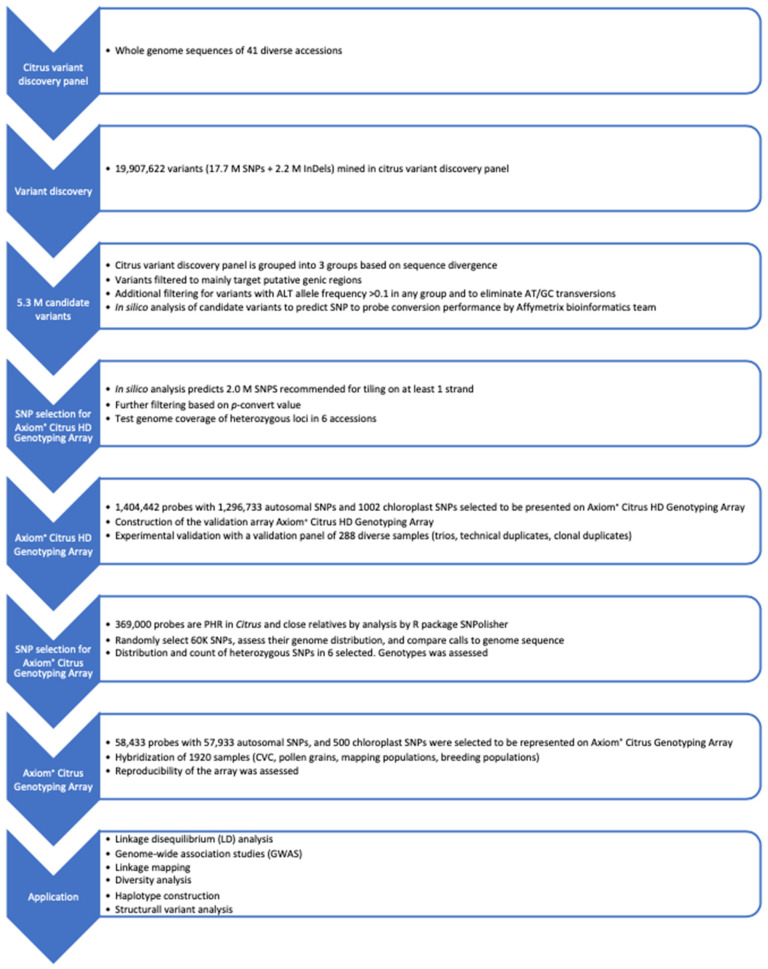
Workflow for citrus variant discovery panel selection, SNP detection, filtering, validation, and final selection of SNPs for Axiom^®^ Citrus HD Genotyping Array and Axiom^®^ Citrus Genotyping Array.

**Table 1 plants-13-00691-t001:** Accessions of *Citrus* and its close relatives in citrus variant discovery panel that were re-sequenced at the UCR Institute of Integrative Genome Biology (IIGB, University of California, Riverside, CA, USA) for designing of Axiom^®^ Citrus HD Genotyping Array and Axiom^®^ Citrus Genotyping Array. These sequences are deposited at NCBI SRA: SRP095606. CRC is accession number in Givaudan Citrus Variety Collection, or for two accessions not yet included in the CVC, the USDA-GRIN database accession code, beginning RSD. Group 1 includes accessions in the mandarin and pummelo groups and complex hybrids, Group 2 includes citrons, and Group 3 includes citrus relatives and papedas.

ID	Variety Name	Genus and Species Name	CRC	Group
MXL	Mexican lime	*Citrus aurantifolia* Swingle	1710	1
JAM	Schaub rough lemon	*Citrus jambhiri* Lush.	3879	1
KSH	Mukakukishu (Kinokuni mandarin)	*Citrus kinokuni* Hort. ex Tanaka	3887	1
LSB	Frost nucellar Lisbon lemon	*Citrus limon* L. Burm. f.	3176	1
TDP	Thong Dee pummelo	*Citrus maxima* (Burm.) Merr	3927	1
CPP	Cariappa CM3 pummelo	*Citrus maxima* (Burm.) Merr.	RSD2013003	1
HUP	Hunan pummelo	*Citrus maxima* (Burm.) Merr.	1225	1
KPP	Kao Phuang pummelo	*Citrus maxima* (Burm.) Merr.	2352	1
MBP	Mato Buntan pummelo	*Citrus maxima* (Burm.) Merr.	3945	1
TAP	Tahitian pummelo	*Citrus maxima* (Burm.) Merr.	3806	1
DUN	Duncan grapefruit	*Citrus paradisi* Macf.	3832	1
TRM	Triumph grapefruit	*Citrus paradisi* Macf.	297	1
CLP	Cleopatra mandarin	*Citrus reshni* Hort. ex Tanaka	3844	1
CSM	Changsha mandarin	*Citrus reticulata* Blanco	3577	1
DNC	Dancy mandarin	*Citrus reticulata* Blanco	3026	1
SCM	Sun Chu Sha Kat mandarin	*Citrus reticulata* Blanco	4003	1
SNK	Sunki mandarin	*Citrus sunki* Hort. ex Tanaka	3143	1
TBM	Tachibana orange	*Citrus tachibana* Tanaka	3150	1
ETH	Ethrog citron	*Citrus medica* L.	3891	2
YUN	Yunnanese citron	*Citrus medica* L.	3798	2
BUD	Buddha’s Hand citron (fingered citron)	*Citrus medica var. sarcodactylis* (Hoola van Nooten) Swingle	3768	2
QIN	Persistent Stigma—OPS citron	*Citrus medica* L.	RSD2012014	2
HYS	Mauritius papeda	*Citrus hystrix* D. C.	3103	3
MIC	Small-fruited papeda	*Citrus micrantha var.microcarpa* Wester	3605	3
ADL	Australian desert lime	*Eremocitrus glauca* (Lindl.) Swingle	3463	3
FOR	Nagami kumquat	*Fortunella margarita* (Lour.) Swingle	3877	3
AFL	Red pulp finger lime	*Microcitrus australasica* var. sanguinia (F. Muell.) Swingle	1484	3
ARL	Australian round lime	*Microcitrus australis* (A. Cunn. Ex Mudie) Swingle	3673	3
FDT	Flying Dragon trifoliate	*Poncirus trifoliata* (L.) Raf.	3330A	3
PON	Pomeroy trifoliate	*Poncirus trifoliata* (L.) Raf.	1717	3

**Table 2 plants-13-00691-t002:** Accessions of *Citrus* and its close relatives in citrus variant discovery panel for which whole genome sequence data were obtained from publicly available sources or our collaborating institution for designing of Axiom^®^ Citrus HD Genotyping Array and Axiom^®^ Citrus Genotyping Array.

Genotype ID	Variety Name	Genus Species Name	Group
BAN *	Banpeiyu	*Citrus maxima* (Burm.) Merr.	1
CHP	Chandler pummelo	*Citrus maxima* (Burm.) Merr.	1
CLM	Clementine mandarin	*Citrus clementina* hort. ex Tanaka	1
KNG *	King mandarin	*Citrus nobilis* Lour.	1
LAP	Low acid pummelo	*Citrus maxima* (Burm.) Merr.	1
PKM	Ponkan mandarin	*Citrus reticulata* Blanco	1
SSO	Sour orange	*Citrus aurantium* L.	1
SWO	Sweet orange	*Citrus sinensis* (L.) Osbeck	1
UNS *	Satsuma mandarin	*Citrus unshiu* Marcovitch	1
WLM	Willowleaf mandarin	*Citrus reticulata* Blanco	1
WMM	W Murcott mandarin	*Citrus reticulata* Blanco	1

* provided by National Institute of Genetics, Japan.

**Table 3 plants-13-00691-t003:** Chromosomal distribution and annotation of SNPs represented on the Axiom^®^ Citrus HD Genotyping Array and the Axiom^®^ Citrus Genotyping Array: (a) Distribution of 1,297,735 SNPs represented on the Axiom^®^ Citrus HD Genotyping Array and 58,433 SNPs represented on the Axiom^®^ Citrus Genotyping Array from alignment to the *C. clementina* v1.0 and *C. sinensis* chloroplast genomes. (b) SNP targets of the Axiom^®^ Citrus HD Genotyping Array and the Axiom^®^ Citrus Genotyping Array against *Phytozome C. clementina* v1.0 reference genome.

**(a)**			
		**Probe Count**	
**Linkage Group**	**Size (Mb)**	**Axiom****^®^** **Citrus HD Genotyping Array**	**Axiom****^®^** **Citrus Genotyping Array**
1	28.94	137,783	6289
2	36.38	171,732	7773
3	51.05	232,654	10,572
4	25.65	132,515	6259
5	43.3	152,556	6665
6	25.61	120,276	5446
7	21.13	115,841	4961
8	25.11	107,769	4608
9	31.41	125,607	5360
Chloroplast	0.16	1002	500
**(b)**			
		**SNP Count**	
**Phytozome Annotation**		**Axiom^®^ Citrus HD Genotyping Array**	**Axiom^®^ Citrus Genotyping Array**
Overlap with START Codons		336	8
Overlap with STOP Codons		38	24
Overlap with Splice Sites		4815	236
Intergenic		665,912	20,144
Gene Body		630,821	38,289
Introns		245,120	14,787
Exons		385,701	23,502
mRNA		630,821	38,289
CDS		291,849	19,550
UTR		93,852	3952
Transcription Start Site		275	9

**Table 4 plants-13-00691-t004:** Summary of SNP classes represented on the Axiom^®^ Citrus HD Genotyping Array and the Axiom^®^ Citrus Genotyping Array. PHR SNPs identified after analysis of 1507 genomic DNA samples with Axiom^®^ Suite default settings.

	Total Probes	Autosomal SNPs	Chloroplast SNPs	SNP Density (SNP/kb)	PHR
Axiom^®^ Citrus HD Genotyping Array	1,404,442	1,296,733	1002	3.9	728,831
Axiom^®^ Citrus Genotyping Array	58,433	57,933	500	0.16	51,296

**Table 5 plants-13-00691-t005:** Accession types of 196 and 871 accessions analyzed with Axiom^®^ Citrus HD Genotyping Array and Axiom^®^ Citrus Genotyping Array.

Species	CVC Type	Axiom^®^ Citrus HD Genotyping Array	Axiom^®^ Citrus Genotyping Array
*C. sinensis* (Hyb)	Blood orange	3	17
*C. sinensis* (Hyb)	Navel orange	6	73
*C. sinensis* (Hyb)	Sweet orange	3	71
*C. sinensis* (Hyb)	Valencia orange	4	26
*C. medica*	Citron	12	20
(*C. reticulata* × *C. medica*)	Rough lemon	6	15
(*C. micrantha* × *C. medica*)	Lime	5	26
(*C. reticulata* × *C. medica*)	Rangpur lime	3	17
*Fortunella* spp.	Kumquat	2	6
(Hyb)	Calamondin	1	3
C. limon (*C. aurantium* × *C. medica*)	Lemon	12	49
*C. limon*	Eureka lemon	2	14
*C. limon*	Lisbon lemon	3	13
*C. reticulata*	Mandarin	35	94
*C. clementina* (Hyb)	Clementine	3	16
*C. unshiu* (Hyb)	Satsuma	3	45
(Hyb)	Tangelo	3	29
(Hyb)	Tangor	5	18
*C. ichangensis*, *C. micrantha*, etc.	Papeda	3	7
*C. maxima*	Pummelo	25	58
(*C. maxima* × *C. sinensis*)	Grapefruit	6	34
*C. aurantium* (*C. maxima* × *C. reticulata*)	Sour orange	5	26
*P. trifoliata*	Trifoliate	6	33
(*C. sinensis* × *P. trifoliata*)	Citrangelo	2	24
*C. paradisi* × *P. trifoliata*)	Citrumelo	1	8
(Hyb)	Hybrids	7	76
	Unknown	30	55

## Data Availability

New whole genome sequence data generated for this study is available at NCBI SRA: SRP095606 and at https://www.citrusgenomedb.org/organism/Citrus/clementina (accessed on 1 January 2024) as a sequence browser. Array annotation details are available from ThermoFisher.

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
