# Peer review of "Development and Assessment of SNP Genotyping Arrays for Citrus and Its Close Relatives"

_plants, 2024, doi:10.3390/plants13050691_

Round 1

Reviewer 1 Report

Comments and Suggestions for Authors

The paper by Hiraoka and colleagues describes the development of two genotyping arrays for citrus and the use of both arrays for genotyping a large germplasm collection. The authors described the advantages of using the SNP arrays, but also the limitations, for example they emphasize that the accuracy decreases for the analysis of citrus relatives. In any case, this study demonstrates the usefulness and reproducibility of both arrays for citrus SNP genotyping and certainly deserves publication.

The introduction should be shortened, avoiding the description of aspects that are not strictly linked to the SNP arrays validation. Results, methods, and discussion that are included at the end of the introduction should be moved to the appropriate sections.

The discussion is based on bibliographic information that is not completely updated. It seems that the most relevant bibliography of the last 5 years regarding citrus phylogeny is missing. Just as an example, Wu et al 2021 Nat Communications indicated that C. ichangensis is an ancestral species, and the results of the present paper should be also discussed in relation to the results of Wu et al 2021.

Other points:

line 28: statistics should be updated

lines 61-62: newer reference genomes have been published in recent years. In my opinion, the most important works of the last 5 years reporting the new reference genomes should be cited.

line 136- 165: this section should be moved in the methods and should be limited to the description of the workflow. All considerations related to the design should be part of the discussion.

lines 217-223: please move this paragraph in the results.

line 382: in the PCA section, the meaning of “clonal samples” is not clear. For example, if the clonal samples were removed, why Eureka lemon is represented 2 and 14 times? This aspect should be better clarified.

In table 3, the CVC type is not clear, it mostly follows an alphabetical order, but in my opinion it should be organized by species. For example, blood oranges, Valencias, Navel oranges, and sweet oranges are all C. sinensis and should stay close to each other. This table should include taxonomical names to facilitate readers.

Comments on the Quality of English Language

.

Reviewer 2 Report

Comments and Suggestions for Authors

Line 28: FAO 2013 and 2014, you can use more recent references (for instance the production is estimated about 200 million tons in 2022 (https://www.fao.org/faostat/en/#data/QCL)

Table 1: Varieties are organized alphabetically by ‘ID’, but it would be better to organize them by ‘Group’ or alphabetically by ‘Genus and Species Name’, for easier reading.

In addition, lemons and limes are classified in group 1 (mandarin and pummelo groups), whereas they could be included in group 2 (citron hybrids). 'Mexican' limes, for example, have no alleles in its genome from mandarins or pummelos. 'Mexican' lime is known as a direct hybrid between Citrus medica (group 2) and Citrus micrantha (group 3) but is included in group 1. Can the authors explain why they choose to put it in this group, and the consequences on the variant filtering (2.3) and the rest of the results? 

Table 2: the "Genus Species Name" of 'Chandler' pummelo and Sour orange are not written in italics.

This article focuses on the use of simple tools for the genetic characterization of diversity. I have the impression that these tools are not necessarily adapted to all breeders' needs, including diversity studies within specific populations between two taxa only, such as Mandarin x Pummelo. Could you explain whether it is or will be possible to develop more specific tools with a narrower spectrum?

This article describes interesting and accessible tools for better understanding the citrus genome, useful both for managing the diversity of collections and for managing recombinant populations as part of breeding programs. These might be not perfect tools but it does meet certain needs even when budgets are limited.

Reviewer 3 Report

Comments and Suggestions for Authors

The authors did comprehensive analysis of SNP array for genotyping and breeding purpose. This group has a long time of research on citrus germplasm and molecular work, with experience on SNP array development based on EST or molecular marker data. This study develped two types of Array based whole genome information, thus are very important for citrus community. The authors did systemic analysis including selection, filtering ,confirmation and reproducibility evalutions. I would recommend pulication of this study upon some minor revisions,

1. What is the relationship between the two arrays? Is the 58K array extracted from the 1.4 M array? Details on any specfic purpose could be helpful for the future users.

2. Bootstrap values maybe better if could be added to the phylogenetic tree.

3. It is somewhat confusing that both Citrus and Citron appear in the same venn diagram such as Fig S1, S2, S3 and S5. In my idea, Citron is belong to Citrus. So the authors may need to revise to avoid misunderstanding.

Reviewer 4 Report

Comments and Suggestions for Authors

The manuscript presented by Hiraoka and colleagues is generally clear, well written and structured. Overall, the results presented represent valuable information to better understand the genetic diversity and relationships of citrus genomes and that could be used for the characterization of citrus germplasm and citrus breeding programs.

In my opinion, the manuscript is suitable for publication, although I have some comments and questions which I include here below:

The intro seems too long. I suggest removing several sentences especially if related to other crops such as in Lines 36-40 and Lines 100-103.

The expression ‘high density’ could be changed with the most correct form ‘high-density’ similarly for ‘high-resolution’, ‘high-fidelity’, ‘well-established’, Please check them in the text.

There is a misuse of the articles 'the' and 'a/an'.

The quality of the figures must be improved.

Supplementary material: check that all species names are in italics;

Although well-detailed, the materials and methods are too long and can be confusing, especially as regards the number of samples analyzed and filtered. I would reduce the text where it is possible to move some sentences in the discussions and better structure the tables, especially the supplementary ones.

Often in materials and methods and in results sections you refer to a different number of samples (254…34..20 samples) for which reference to a well-structured additional table would be necessary.

Line 28: remove the comma after (FAO, 2014);

Line 32: change ‘has’ with ‘have’;

Line 65: add ‘an’ before ‘average’;

Line 66: add ‘the’ before the ‘availability’;

Line 82: add ‘the’ before ‘release’;

Line 99: remove ‘the’ before ‘recent’;

Line 136: ‘Design of Genome-Wide SNP … Genotyping Array’, it would have been appropriate to place it in the section materials and methods;

Line 158: add ‘the’ before ‘performance’;

Line 168: ‘Leaf samples’, indicate the table containing information on the samples used

Line 189: ’30 accessions’, the reference to the table 1 is missing;

Line 192: ’11 accessions’, the reference to the table is missing;

Table 2: check that all species names are in italics;

Line 201: explain why clementine was chosen and not sweet orange as a reference genome considering that you used the sweet orange chloroplast genome for chloroplast variant discovery and that the latest sweet orange reference genomes provide a completer and more accurate overview of the whole genome. 

Line 200: please provide more details on the pipeline used;

Line 202: ‘(Bausher et al., 2006)’, please add ref in the ‘References’ section;

Line 220: also here more details on the filtering process will help reader in understanding what has been done;

Line 223: ‘(Table S1)’ I would combine the results shown in table s1 with tables 1 and 2, maybe you could remove you may remove column ‘CRC’;

Line 259: add ‘s’ to ‘candidate’;

Line 313: 277 genomic DNA samples (Table S2), the caption shows 232 samples instead, also, I suggest to add in the table a column with the species and group of each sample;

Line 323: ‘Variant Filtering Part 2’, I think it is not necessary to use the bulleted list (22.,23.,24.) in this paragraph;

Line 326: ‘Very distant relative, polyploid and synthetic polyploid (mixed DNA) samples were removed from the set of 277 genomic DNA samples’, has the ploidy level been verified? If so how, indicate in the text or in the additional material such as table 2s the samples removed;

Line 333: ‘254 genomic DNA samples’, I mean the number of samples after removal of polyploid samples;

Line 347: remove ‘s’ in ‘numbers’;

Line 372: add ‘the’ before ‘reproducibility’;

Line 379: add ‘the’ before ‘reproducibility’;

Lines 385-389: ‘A total of 196 accessions….duplicate samples (Table 3).’  The sentence is too long. Please rephrase for clarity;

Lines 398-411: ’26 selected accessions…87-33 accessions’, indicate which, maybe in one of the tabs of the additional materials;

Line 416: ‘in order for C. micrantha var. microcarpa to be included in the phylogenetic analysis’ change it with ‘to include C. micrantha var. macrocarpa in the phylogenetic analysis’;

Line 437: ‘26 selected CVC accessions including 25 accessions’, indicate which, maybe in one of the tabs of the additional materials;

Line 458: ‘Axiom® Citrus HD Genotyping Array is comprised of Axiom® Citrus HD Genotyping Array AX1 and Axiom® Citrus HD Genotyping Array AX2’, would be 15AX1 and 15AX2…explain why two arrays AX1 and AX2;

Line 462: ‘SNPs from (polymorphic in)’, I don’t understand what you mean by polymorphic in, rephrase for clarity;

Line 465: change ‘. (Fig S3)’ with ‘(Figure S3).’;

Lines 469-471: ‘254 genomic DNA samples, of which 232 were unique … of which 220 samples (86.61%) passed and 34 samples failed’, please indicate the table, a well-structured additional table of 254 samples would be needed for both materials and methods and results;

Lines 497-500: ’34 samples…20 samples…10 samples’, indicate which and the table;

Line 506: Table 5, Axiom Citrus 56 AX?  You didn't discuss the 56 AX, should be 15AX1 and 15AX2, please clarify;

Line 510: Figures a-b, it would be appropriate to add to the graph the percentages for each category;

Line 547: Figure S6, please improve the quality of the figure, it is difficult to understand;

Line 568: add ‘the’ before ‘highest’;

Line 569: add ‘the’ before ‘lowest’;

Lines 592-594: ‘In addition, average….(Table S1).’ Add ‘the’ before ‘average’, remove the point before (Table S1), add a comma after ‘heterozygosity’ and ‘Genotyping Array’ or rephrase for clarity;

Line 688: ‘(Figure 9a) and sequence data (Figure 9b)’, there is no distinction between Figure 9a and Figure9b but only Figure 9;

Line 639: add ‘the’ before ‘percent’;

Line 727: add ‘a’ before ‘fairly’;

Line 791: add ‘the’ before ‘great’;

Line 841: add ‘an’ before ‘insufficient’;

Line 862; change ‘It is likely that increased marker density was able to’ with ‘Increased marker density was likely able to’;

Line 933: remove the second point after ‘by bold lines.’;

Line 946: add ‘the’ before ‘diversity’;

Line 947: use ‘various’ or ‘different’;

Line 952: add ‘the’ before ‘detection’;

Line 953: add 'the’ before 'parentage’;

Line 954: add 'the’ before 'correct’;

Line 959: add ‘the’ before ‘annotation’.

Comments on the Quality of English Language

The quality of the English language is good, although minor editing is required.

Round 2

Reviewer 1 Report

Comments and Suggestions for Authors

The authors made the requested changes, and the paper can be accepted in the present form